New craniodental remains of Thylacinus potens (Dasyuromorphia: Thylacinidae), a carnivorous marsupial from the late Miocene Alcoota Local Fauna of central Australia

Yates Adam M. adamm.yates@nt.gov.au
Museums and Art Galleries of the Northern Territory , Museum of Central Australia, Alice Springs, Northern Territory , Australia
Farke Andrew
Electronic publication date: 2014 Aug 28
Publication date: 2014
Volume: 2
Electronic Location ID: e547
Received 2014 Mar 26; Accepted 2014 Aug 7
Copyright: © 2014 Yates
Copyright year: 2014
Copyright holder: Yates
License: This is an open access article distributed under the terms of the Creative Commons Attribution License, which permits unrestricted use, distribution, reproduction and adaptation in any medium and for any purpose provided that it is properly attributed. For attribution, the original author(s), title, publication source (PeerJ) and either DOI or URL of the article must be cited.
License URL: https://creativecommons.org/licenses/by/4.0/

Keywords: Miocene, Australia, Thylacinidae, Thylacinus potens, Alcoota, Body size

Funding: The author received no funding for this work.

==============================
New craniodental specimens that are referrable to the thylacinid marsupial, Thylacinus potens, are described from the late Miocene Alcoota Local Fauna of the Northern Territory, Australia. The remains include a largely complete maxilla and dentary, showing for the first time the anterior dentition of the dentary. The new remains indicate that Th. potens was a more variable species than previously recognised. The dentary, in particular, is more gracile, than other specimens referred to this species. A revised apomorphy-based diagnosis of Th. potens that takes this variability into account is presented. A cladistic analysis supports previous analyses that placed Th. potens in a derived position within Thylacinidae, close to the modern Th. cynocephalus. New estimations of body size are made using published regressions of dental measurements of dasyuromorphians as well as by assuming geometric similitude with Th. cynocephalus. All methods produce body mass estimates in excess of 35 kg.

Introduction

The recently extinct ‘Tasmanian tiger’ (Thylacinus cynocephalus) was Australia’s largest surviving mammalian carnivore at the time of European settlement. It belongs to a family of marsupial carnivores, the Thylacinidae, whose fossil record extends back to the late Oligocene (approximately 24 ma) (Wroe, 2003). Molecular phylogenetics based on surviving Th. cynocephalus tissues have revealed conclusively that the Thylacinidae was part of Dasyuromorphia, a clade of australodelphian marsupials that include the surviving Australo-Papuan marsupial carnivores, the Dasyuridae, and the numbat (Myrmecobius fasciatus), an ant-eating specialist (Krajewski, Buckley & Westerman, 1997; Beck, 2008; Miller et al., 2009).

Many thylacinids had body sizes equal to, or larger than the largest dasyurids, although there is substantial overlap in body size between the two families (Wroe, 2001). The very largest thylacinids belong to the derived genus Thylacinus and one of the largest known species of this genus is the late Miocene Th. potens Woodburne, 1967 from the Alcoota local fauna, possibly reaching a body weight close to 40 kg (Wroe, 2001). Th. potens lived at an important time in the history of thylacinids, when the great diversity of late Oligocene and early-middle Miocene small-bodied thylacinids were becoming extinct and Thylacinus species were evolving larger size, presumably as a specialisation towards predation on large-bodied vertebrate prey. Unfortunately our anatomical and palaeobiological knowledge of Th. potens remains poor due to its frustratingly meagre fossil record. The original hypodigm of Th. potens consisted of the holotype palate, two dentary fragments, some teeth and a few postcranial elements from the hind foot (Woodburne, 1967). Since that time, scant material has been added, despite extensive annual excavations at Alcoota carried out over a 27 year period by a joint team from Flinders University (FU) and the Museum and Art Gallery of the Northern Territory (MAGNT). However, the 2013 expedition saw the discovery of the first substantial craniodental remains of this species to be recovered since Woodburne’s initial excavation of the Alcoota Local Fauna in 1962–63.

These new specimens expand our knowledge of the anatomy of this species and its range of variation. As a result of this new information the diagnosis of the species is revised. With more complete specimens at hand, new estimates of the size of Th. potens are also calculated.

Geological setting

The Alcoota Local Fauna is known from a dense bone bed in the lower part of the Waite Formation, cropping out on Alcoota Station, 110 km NE of Alice Springs in south central Northern Territory (Woodburne, 1967). The Waite Formation is a late Cenozoic sequence of fluviatile silts, sands and minor limestone beds filling the Waite Basin, a small intermontane basin, surrounded by crystalline rocks of the Arunta Block (Woodburne, 1967). The bone bed covers an area of approximately 25,000 m2, although its density and thickness varies considerably within that area (Megirian, 2000). The bonebed usually lies 90 cm below the present soil surface, underneath a reddish, weathered horizon (Murray & Megirian, 1992). The bulk of the known fossil material has been obtained from three pits: Paine Quarry, South Pit and Main Pit. Paine Quarry was excavated by Woodburne and colleagues from 1962 to 1963, and produced the original Th. potens material that was described by Woodburne (1967) when he erected the species. The pit was presumably backfilled at the end of Woodburne’s field investigations in 1963. South Pit and Main Pit were opened by a team from FU and MAGNT in the mid 1980s and have been kept open and have been more or less continually excavated up to the present. The precise location of Paine Quarry in relation to the FU-MAGNT pits has always been uncertain but recent work matching old, long-lived trees to those present in Woodburne’s original field photographs indicates that it lay immediately west of the present day South Pit (Fig. 1).

Figure 1 Locality map.

Map of north–west corner of Alcoota Fossil Reserve showing the principal excavation sites of the Alcoota Local Fauna.

Based on stage of evolution correlation it is thought that the fauna is late Miocene in age (Stirton, Woodburne & Plane, 1967; Murray & Megirian, 1992), and lies somewhere between 5 and 12 million years old (Megirian et al., 2010). The fauna is dominated by large browsing herbivores, both mammalian and avian. Mammalian carnivores are exceptionally rare and restricted to just three known species: Th. potens, Tyarrpecinus rothi and Wakaleo alcootaensis (Woodburne, 1967; Murray & Megirian, 2000; Archer & Rich, 1982). Of these, Ty. rothi is a small thylacinid, weighing about 5 kg (Wroe, 2001) that is known only from a single fragmentary specimen (Murray & Megirian, 2000).W. alcootaensis is a leopard-sized thylacoleonid weighing about 35 kg that is known from even less material than Th. potens. Apart from the specimens of Th. potens described by Woodburne (1967), FU-MAGNT excavations in Main Pit have produced a few postcranial elements, a canine crown, a largely uninformative molar fragment and a single heavily worn and broken molar.

During the 2013 field season a new pit was opened between the Main Pit and South Pit, at the same stratigraphic height as these two quarries (Fig. 1). This new pit, named ‘Shattered Dreams’, proved to be exceptionally densely packed with fragmented bones, interspersed with occasional complete, or near complete specimens. Not only was the volume of fossil bone extraordinarily high but the diversity was also high with almost all of the known taxa from the Alcoota Local Fauna recovered from an excavated area of less than two square meters. Among these specimens are postcranial elements as well as upper and lower jaw bones bearing teeth that are referrable to Th. potens. In addition an isolated premolar was discovered in South Pit. These are the first substantial craniodental remains of this species to be recovered since Woodburne’s initial excavation of the Alcoota local fauna.

Methods

Terminology

Serial designation of the cheek dentition follows Flower (1867) and Luckett (1993). Standard nomenclature for mammalian tooth cusp anatomy is followed. Anterior and posterior are used as anatomical directions in the description of the dentition (replacing mesial and distal, respectively). This is done to bring the description into line with the rest of the descriptive literature on thylacinids. As the jaws of thylacinids are elongate and the dental arcades are rather straight and anteroposteriorly oriented, anterior and posterior are synonymous with the dental directional terms mesial (toward the symphysis) and distal (away from the symphysis), at least for the canine and post canine dentition.

Institutional Abbreviations are as follows: CPC, Commonwealth Palaeontological Collection, Geoscience Australia, Canberra; NTM, Museum and Art Gallery of the Northern Territory, Darwin and Alice Springs; SAM, South Australian Museum, Adelaide; UCMP, Museum of Paleontology, University of California, Berkeley.

Size estimation

Two methods were used to calculate the body mass of the new Th. potens specimens: NTM P4326 and NTM P4327. Firstly, some of Myers’ (2001) regressions were selected. These regressions were derived to predict body mass of marsupials from a series of craniodental measurements. Three such regressions were used, all derived from the restricted dasyuromorphian dataset (Myers, 2001, Table 4). The regression for lower molar tooth row was used to estimate the mass of the dentary specimen (NTM P4327) while the regressions for upper molar tooth row length and width of M2 was used for the maxilla specimen (NTM P4326). Unfortunately M1 of NTM P4326 is badly damaged and isolated from the rest of the molar row. Consequently the length of the upper molar tooth row could only be estimated, hence the use of the second, less accurate predictive variable. Myers (2001) applied a smearing estimate (Duan, 1983) to his predicted body mass values to correct for logarithmic transformation bias that results when the predicted value is detransformed. The same smearing estimates are applied here.

The second method used to estimate body mass follows that of Wroe (2001), who assumed geometric similitude between large-bodied species of Thylacinus and obtained a scaling factor by comparing measurements of the fossils with the average of the same measurement from Th. cynocephalus. In this study the measurements used to obtain the scaling factors were lower molar tooth row length for the dentary specimen (NTM P4327) and the combined length of M2–4 for the maxilla specimen (NTM P4326). Average values for Th. cynocephalus were obtained from Wroe (2001, Table 4) and Woodburne (1967, Table 1). An average mass of 29.5 kg for Th. cynocephalus (Paddle, 2000) was used to scale body mass.

Cladistic analysis

Several cladistic analyses of thylacinid and dasyuromorphian phylogeny have been attempted. However, no published character-taxon matrix includes all available informative characters and all thylacinid taxa described to date. Therefore a new matrix was assembled by combining data from previous analyses with the addition of some new character scores for Th. potens and Th. megiriani. The ingroup was restricted to Thylacinidae and characters that were uninformative within the restricted ingroup were excluded. Two dasyurids (Barinya wangala and Antechinus flavipes) were employed as outgroup taxa. New character states for Th. potens were taken from the specimens described in this article while new character states for Th. megiriani were taken from undescribed lower jaw specimens held in the NTM collections (NTM P4376, 4377). Polymorphisms were treated as uncertainty. Terminal taxa used and their sources of character data are given in Table 1. Body mass was employed as an ordered multistate character. This is not a common practice because there is a belief that body size is too plastic to be useful for cladistics analysis, and that such phenomena as sexual dimorphism would confound its use. However the same could be said of a great many characters routinely employed in cladistics analysis. Body size is included here because it does display a high degree of heritability and therefore contains phylogenetic signal and its exclusion would violate the principal of total evidence. Furthermore body size is one of relatively few characteristics that varies between the larger derived species of Thylacinus. Specifically the exceptional size of both Th. potens and Th. megiriani is a potential synapomorphy linking these two species as sister taxa. Such an arrangement has significant impacts for the reconstruction of thylacinid evolution in the late Cenozoic and deserves to be tested as thoroughly as possible. Characters (Appendix 1) were taken from Murray (1997), Muirhead & Wroe (1998), Wroe & Musser (2001) and Murray & Megirian (2006) with some modification. A few novel characters were added.

Table 1 Terminal taxa.

Terminal taxa used in the cladistic analysis and their sources of character data (literature and specimens).

Taxon	Sources	
Dasyuridae	Wroe (1999) (Barinya wangala); NTM U7542 (Dasyurus maculatus)	
Muribacinus gadiyuli	Wroe (1996)	
Badjcinus turnbulli	Muirhead & Wroe (1998)	
Ngamalacinus timmulvaneyi	Muirhead (1997)	
Maximucinus muirheadae	Wroe (2001)	
Mutpuracinus archibaldi	Murray & Megirian (2000), Murray & Megirian (2006); NTM P907-3; NTM P9612-5	
Nimbacinus dicksoni	Muirhead & Archer (1990), Wroe & Musser (2001)	
Nimbacinus richi	Murray & Megirian (2000); NTM P9612-4; NTM P9973-11	
Wabulacinus ridei	Muirhead (1997)	
Tyarrpecinus rothi	Murray & Megirian (2000); NTM P98211	
Thylacinus macknessi	Muirhead (1992), Muirhead & Gillespie (1995)	
Thylacinus potens	Woodburne (1967); CPC 6746(c); NTM P4326; NTM P4327	
Thylacinus megiriani	Murray (1997); NTM P4376; NTM P4377; NTM P9618	
Thylacinus cynocephalus	Murray & Megirian (2006); SAM M95, SAM M665/001, SAM M922, SAM M1952-56, SAM M1959-60.	

The resulting matrix was subjected to a maximum parsimony analysis in PAUP 4.0b (Swofford, 2002) using the following settings: heuristic search; random addition sequence with 500 replicates; and TBR branch-swapping algorithm. The strength of the internal nodes was tested with a bootstrap analysis (1000 bootstrap replicates, heuristic searching with 50 addition sequence replicates).

Graphical representation of the common cladistic information of the most parsimonious trees (MPTs) was achieved by a posteriori pruning of labile taxa, i.e., a reduced cladistic consensus tree (Wilkinson, 1994) was produced. Due to the small number of MPTs, this was achievable by visual inspection of the most parsimonious trees which revealed that all of the loss of resolution in the strict consensus tree was caused by the variable position of a single taxon, Maximucinus muirheadae. This was then pruned from the MPTs, to produce the reduced cladistics consensus tree.

Photography

Monochrome images were prepared by coating the specimens in ammonium chloride and focus-stacking multiple images taken at different focal depths. The interpretive drawings were made from earlier, lower-quality photographs. As a result the drawings do not precisely match the images presented alongside them, however the discrepancies are quite minor and of no consequence.

Systematic Palaeontology

Dasyuromorphia Gill, 1872

Thylacinidae Bonaparte, 1838

Thylacinus potens Woodburne, 1967

Holotype. CPC 6746, “a palatal fragment with RM2–M4 and LP2–M2, preserved. Other teeth are represented by roots and alveoli” (Woodburne, 1967, pg. 20).

Type locality. Paine Quarry, Alcoota Station.

New material. NTM P4326, right maxilla in 2 parts, with complete P2–3, M2–4, and fragments of P1 and M1 from Shattered Dreams (Figs. 2–9); NTM P4332, isolated left P3 from South Pit (Fig. 10); NTM P4379, maxillary fragment with broken and worn right M2 from Main Pit (Fig. 11); NTM P4327, left dentary with P2–3, M1–4 and root fragments of P1 from Shattered Dreams (Figs. 12–15); NTM P4461, crown of right C1 from Main Pit (Fig. 16); NTM P4516, fragment of right upper molar, possibly M1, from an unrecorded site of the Alcoota Local Fauna.

Figure 2 Whole maxilla.

Thylacinus potens. NTM P4326, right maxilla. (A) lateral view, (B) ventral view, (C) reconstruction of palate by mirror imaging the right side. Abbreviations: Ca, canine alveolus; if, incisive foramen; iof, infraorbital foramen; M1–4, molars 1–4; P1–3, premolars 1–3; pf, palatine fenestra; sym, symphyseal surface. Scale bars = 50 mm.

Emended diagnosis. Thylacinid distinguished by the following unambiguous autapomorphies: long axis of P1 anterobuccally oriented in adults; anterior width of the first upper molar greater than its anterior–posterior length; reduced palatal fenestrae approximately one third the length of the upper molar tooth row; absence of a diastema between P1 and P2; P2 longer than P3 and M1. The following ambiguous synapomorphies serve to distinguish Th. potens from Th. cynocephalus (and most fossil thylacinids): ventrally facing sulcus forming the ventral border of the root of the zygomatic arch on the maxilla; P2 longer than M1.

Description

Maxilla

The maxilla (NTM P4326) includes the tall side wall of the rostrum that is absent in the holotype. The height of the maxilla above the anterior edge of P3 is 44.4 mm which is 67.3% of the distance from the anterior margin of the canine to the posterior margin of P3 or approximately 42% of the total length of the cheek tooth row. These proportions lie with the range of Th. cynocephalus (Table 2). Furthermore, the anterodorsal margin of the maxilla rises from the level of the canine to the level of P3 at an angle of 32° (Fig. 4A), which matches the angle seen in Th. cynocephalus. These observations indicate that the snout of Th. potens was probably not proportionately shorter or deeper than that of Th. cynocephalus and that the crowding of the premolar teeth seen in this species is more likely to be the result of relative enlargement of these teeth as opposed to relative shortening of the jaw (Fig. 3). In anterior view the lateral wall of the maxilla slopes dorsomedially, indicating that the rostrum was triangular in cross section.

Table 2 Cranial measurements and ratio for Thylacinus species.

Selected cranial measurements and ratio of Thylacinus potens and Th. cynocephalaus. Measurements in mm, ∼ indicates an approximation due to damage, ∗ indicates a transverse measurement obtained by doubling the distance from the landmark to the midline. Measurements for Th. cynocephalus were obtained from a sample of nine adult specimens held at SAM.

	MH	C-P3	MH/C-P3	P1–P1	DD	
Th. potens						
NTM P4326	44.4	∼66.0	67.3%	19.2*		
NTM P4327					30.3	
UCMP 66206					37.0	
Th. cynocephalus						
Mean	34.6	52.2	66.2%	22.1	27.2	
Range	28.8–40.7	45.8–58.52	63.9%–70.5%	18.9–23.9	22.1–31.0	
Notes.

MH vertical height of the maxilla above the mesial end of P3

C-P3 the distance between the mesial margin of the upper canine and the distal margin of P3

MH/C-P3 ratio of maxilla height to canine-P3 distance

P1–P1 transverse distance between the left and right lingual sides of the distal roots of each P1

DD depth of the horizontal ramus of the dentary measured at the level of the mesial end of M4

Figure 3 Comparison of Thylacinus maxillae.

Comparison of the maxilla of Thylacinus cynocephalus and Th. potens. (A) left maxilla of Th. cynocephalus in lateral view, (B) right maxilla (reversed for comparison) of Th. potens in lateral view. Both drawings scaled to the same maxillary length for comparison. (A) redrawn from Murray & Megirian (2006, Appendix 1, Fig. 1), (B) based on NTM P4326.

Figure 4 Photographs of anterior part of maxilla.

Thylacinus potens. NTM P4326, detail of the anterior fragment of the right maxilla. (A) lateral view, (B) ventral view, (C) medial view. Scale bar = 20 mm.

Ventrally the palatal shelf of the maxilla is complete between the canine and P2. It indicates that the anterior palate in this region was flat and narrow and was located just a couple of millimetres above the lingual alveolar margins. Doubling the distance from the lingual side of the posterior root of P1 to the midline symphysis indicates that the total width of the palate between the posterior roots of P1 is 19.2 mm, distance almost identical to that of the holotype specimen. This is unusually narrow in comparison to Th. cynocephalus, and lies at the small end of the range displayed by that species (Table 2), indicating that Th. potens may have had a relatively narrow anterior end of the snout (Fig. 2C). A small notch at the anterior end of the preserved portion of the palate is the posterior end of the incisive foramen. It indicates that in this specimen the posterior terminations of these foramina lay between the anterior ends of the canine alveoli, well anterior their position in Th. cynocephalus where they terminate between the canine and P1. The palate behind the incisive foramen is simple and flat without the depression or low transverse ridge seen in the holotype.

Figure 5 Drawings of anterior part of maxilla.

Thylacinus potens. NTM P4326, anterior fragment of the right maxilla, interpretive drawings of the photographs in Fig. 4. (A) lateral view, (B) ventral view, (C) medial view. Abbreviations: an, articulation surface for nasal; apm, articulation surface for premaxilla; ar, anterior root; ara, alveolus of the anterior root; arP1–3, anterior roots of premolars 1–3; bh, basal heel; Ca, canine alveolus; if, incisive foramen; mp, palatal shelf of the maxilla; por, posterior root; porP1–3; distal root of premolars 1–3; pr, protocone; sym, symphyseal surface. Grey fill represent areas of adherent matrix, areas hatched with continuous horizontal lines represent broken bone and tooth surfaces. Scale bar = 30 mm.

The posterior maxillary fragment bears the ventral floor of the infraorbital canal on its dorsal surface (Figs. 7C and 8C). From the extent of the broken medial and lateral walls of this canal it is apparent that the lateral opening of this canal, the infraorbital foramen, lay above M2 (Fig. 6B), approximately level with its midlength, as it does in Th. cynocephalus. The lateral margin of the canal dwindles anteriorly to a thin ridge that terminates posterior to the contact between M1 and M2, indicating that the infraorbital foramen could not have occupied the anterior position that it does in the holotype of Th. potens. As in the holotype there is a well-developed, ventrally-facing sulcus incised into the posterior lateral surface of the maxilla, forming the ventral margin of the anterior root of the zygomatic arch (Figs. 7A, 7D, 8A and 8D). Also resembling the holotype is a well-developed pit on the palate between the protocone alveoli of M3 and M4. There is a much shallower and less distinct fossa in the analogous position between M2 and M3. Two short sections of natural edge are present along the largely broken medial margin of the maxillary shelf level with M1 and M2. These represent part of the lateral margin of the palatal vacuity. They indicate that the vacuity lay just 5.8 mm from the protocone of M2, however there is not enough edge preserved to determine the relative size of the vacuity. Neither is the maxillary shelf complete enough to determine the posterior width of the palate.

Figure 6 Posterior part of maxilla in lateral and medial views.

Thylacinus potens. NTM P4326, detail of the posterior fragment of the right maxilla. (A) lateral view, (B) interpretive drawing of A, (C) medial view, (D) interpretive drawing of C. Abbreviations: aj, articulation surface for jugal; ap, articulation surface for palatine; ioc, infraorbital canal; me, metacone; mpf, margin of the palatal fenestra; ms, metastyle; pa, paracone; pr, protocone; ps, parastyle; sB, stylar cusp B; sD, stylar cusp D. Arrow indicates the level of the posterior margin of the infraorbital foramen. Grey fill represent areas of adherent matrix, areas hatched with continuous horizontal lines represent broken bone and tooth surfaces. Scale bar = 30 mm.

Figure 7 Photographs of posterior part of maxilla in ventral, anterior, dorsal and posterior views.

Thylacinus potens. NTM P4326, detail of the posterior fragment of the right maxilla. (A) ventral view, (B) anterior view, (C) dorsal view, (D) posterior view. Scale bar = 30 mm.

Figure 8 Drawings of posterior part of maxilla in ventral, anterior, dorsal and posterior views.

Thylacinus potens. NTM P4326, posterior fragment of the right maxilla, interpretive drawings of the photographs in Fig. 7. (A) ventral view, (B) anterior view, (C) dorsal view, (D) posterior view. Abbreviations: aj, articulation surface for jugal; ap, articulation surface for palatine; ioc, infraorbital canal; me, metacone; mpf, margin of the palatal fenestra; ms, metastyle; pa, paracone; pr, protocone; ps, parastyle; sB, stylar cusp B; sD, stylar cusp D; vs, ventral sulcus. Grey fill represent areas of adherent matrix, areas hatched with continuous horizontal lines represent broken bone and tooth surfaces. Scale bar = 30 mm.

Maxillary dentition

The canine alveolus indicates a large, buccolingually compressed and anteriorly directed canine. As in the holotype, the premolars are significantly larger than those of Th. cynocephalus (Table 3). The double-rooted P1 is represented by its alveolus and the posterior root bearing a small remnant of the crown. Although none of the crown morphology can be determined it is apparent from the alveolus that the long axis of the tooth in occlusal view is canted buccoanteriorly relative to the long axis of the canine and the succeeding premolars (Figs. 4B and 5B). The out-turned anterior margin of the P1 alveolus lies buccal to the posterior margin of the canine alveolus. In lateral view the two margins draw level with each other so that there is no diastema between the two teeth.

Table 3 Measurements of upper premolars of Thylacinus potens and Th. cynocephalus.

Data for CPC 6746 and Th. cynocephalus are taken from Woodburne (1967). Measurements for Th. cynocephalus are mean values taken from a sample of six specimens. Measurements are in mm, ∼ indicates an approximation due to damage.

	P1L	P1W	P2L	P2W	P3L	P3W	
Th. potens							
NTM P4326	∼13.3	4.8	14.2	6.7	15.8	8.7	
NTM P4332	–	–	–	–	16.7	9.4	
CPC 6746	–	–	12.4	5.5	16.0	8.8	
Th. cynocephalus							
Mean	6.2	3.3	8.3	3.8	10.6	5.0	
Notes.

L mesiodistal length

W maximum buccolingual width

A short diastema of 3.0 mm separates P1 from P2. P2, like the other premolars, is a anteroposteriorly elongate and buccolingually compressed, double-rooted tooth. It is worn to such a degree that the crown is reduced to a low, bluntly rounded, mound-like structure with no discernable cusps. The long axis of the crown in occlusal view is aligned with that of P3. The tooth is distinctly wider at its posterior end than at its anterior end. Although the crowns of P2 and P3 do not contact each other their respective alveoli are in contact and there is no diastema between them (Figs. 4B and 5B).

P3 is also heavily worn although the large central protocone remains discernable and distinct from the posterior heel of the crown. NTM P4332 is an isolated P3 in a less worn state (Fig. 10). It shows that the protocone formed a tall conical spike with its apex directed slightly posteriorly. The protocone has a rounded cross-section with no cristae extending up either the anterior or posterior sides. The anterior face of the protocone forms a surface that continues to the base of the crown without any change in slope or development of anterior bulges or cuspules. There is a suggestion of a basal bulge on the anterior side of the P3 of NTM P4326 but this is an artefact produced by a wear facet on the anterior face of the protocone. The anteroventrally sloping anterior profile continues in a straight line onto the upper part of the root before curving posteriorly, creating a distinctly rounded profile in lateral view. The posterior profile of the crown has a distinct basal heel, separated from the posterior margin of the protocone by an inflection. A second rounded posterior cuspule arises from the posterolingual surface of the crown base. This cuspule is positioned basal to the level of the posterior heel. The posterior root extends straight down and is not curved like the anterior root. Due to the breakage of NTM P4332 it is not possible to determine if there was a diastema between P3 and M1.

Figure 9 Close-up of upper molar dentition.

Thylacinus potens. NTM P4326, detail of upper molar tooth row in occlusal view. (A) photograph, (B) interpretive drawing of A. Abbreviations: ef, ectoflexus; me, metacone; ms, metastyle; pa, paracone; pc, precingulum; pmc, postmetacrista; ppc, postparacrista; pprc, postprotocrista; pr, protocone; prpc, preparacrista; ps, parastyle; sB, stylar cup B; sD, stylar cusp D. Areas hatched with continuous horizontal lines represent broken tooth surfaces, areas hatched with discontinuous lines represent wear surfaces. Scale bar = 20 mm.

Figure 10 Upper premolar.

Thylacinus potens. NTM P4332, isolated left P3. (A) buccal view, (B) occlusal view, (C) lingual view, (D) interpretive drawing of A, (E) interpretive drawing of B, (F) interpretive drawing of C. Abbreviations: ar, anterior root; bh, basal heel; plcl, posterior lingual cuspule; por, posterior root; pr, protocone. Grey fill represent areas of adherent matrix, areas hatched with continuous horizontal lines represent broken tooth surfaces, areas hatched with discontinuous lines represent wear surfaces. Scale bar = 10 mm.

A fragment of M1 was recovered from the gap between the two maxillary fragments of NTM P4326. It includes the protocone and the anterobuccal corner of the tooth and their respective roots. The protocone is set lower than the paracone and stylar cusp B, as it is in the other molars. It is worn flat in anterior view and is rounded in occlusal view. A weakly developed precingulum extends along the anterior margin from the linguoanterior corner of the protocone to stylar cusp B. A distinct flexus in the middle of the anterior margin divides the precingulum into two parts, one bordering the protocone, the other the paracone/stylar cusp B complex. The paracone is a low, rounded tubercle. The lingual side of the paracone is somewhat ‘tented’ with a rounded ridge sloping down from the apex of the paracone to the valley that divides it from the protocone. The precingulum terminates in a poorly developed stylar cusp B. This cusp is no more than a low rounded bulge situated on the buccoanterior side of the paracone.

M2 is complete although somewhat worn. In occlusal view there is a shallow ectoflexus between stylar cusps B and D, at about 40% of the length of the buccal margin from the anterior end (Fig. 9), unlike the holotype which bears a deep ectoflexus similar to that of M3. The anterior and posterolingual margins bear weakly developed constrictions in occlusal view, between the protocone and the buccal cusps. A narrow, rudimentary precingulum slopes basally from the anterior side of stylar cusp B to a point near the base of the anterior side of the crown, below the paracone. The low talon is broadly U-shaped in occlusal view and bears a slightly raised protocone on its lingual apex. The preprotocrista, if it was ever present, has been obliterated by a large steeply angled wear facet occupying the anterior face of the talon. The weakly developed postprotocrista extends horizontally along the posterolingual margin of the talon, terminating immediately lingual to the minute metaconule. The metaconule forms a barely-raised, narrow, semi-lunate shelf around the lingual base of the metacone. The paracone has been worn flat, though its smaller base indicates that it was almost certainly subordinate to the metacone before wear. No paracristae remain but the premetacrista component of the centrocrista can be seen extending down the anterior side of the metacone to the worn base of the paracone. The metacone is the largest cusp of the tooth, its apex has been worn off forming an oblique posterobuccally facing wear facet. The postmetacrista is more of a sharp edge than a raised crest as it is in Th. cynocephalus. It curves downwards and buccoventrally from the distolingual edge of the metacone terminating at the posterobuccal corner of the tooth, the posterior end of the metastylar wing. This corner is flat and there is no trace of a raised metastyle. The buccal margin of the stylar shelf forms a raised crest that is higher than the postmetacrista, consequently the metastylar basin faces lingually as opposed to buccally as it does in Th. cynocephalus. The stylar crest rises as it extends anteriorly from the metastylar corner, ending in a well-developed stylar cusp D, which forms a anteroposteriorly elongate and buccolingually compressed cusp. It lies buccally and slightly posterior to the metacone and is the second tallest cusp of the tooth in its present state of wear. A saddle connects stylar cusp D with the metacone that together with the metacone, separates the anterior end of the metastylar basin from the rest of the crown. The stylar crest is terminated by the ectoflexus, anterior to stylar cusp D with the latter being linked to stylar cusp B by a low, rounded saddle. A very small but deep pit is located immediately lingual to this saddle, between stylar cusp D, the metacone and stylar cusp B.

The M3 of NTM P4326 is 2–5% larger than M2 in all measured dimensions (Table 4), unlike the holotype specimen (Woodburne, 1967). In occlusal view there is a well-developed ectoflexus on the buccal margin between stylar cusps B and D, closer to the midlength of the tooth than in M2. The flexure is more strongly developed than in M2, nonetheless it is not as deep as in the M3 of the holotype or the isolated M3 described by Woodburne (1967). The anterior and posterolingual margins bear weakly developed constrictions between the protocone and the buccal cusps. A weakly distinct precingulum extends from the apex of stylar cusp B to a point at the base of the crown adjacent to the anterior constriction. The talon is narrower and more triangular in occlusal view than in M2. The anterior face of the talon curves smoothly onto the occlusal surface of the talon with no preprotocrista defining its margin. The postprotocrista slopes gently down from the apex of the protocone, along the posterolingual margin of the talon. As it approaches the posterolingual surface of the metacone the crista curves sharply towards the base of the crown and a short groove separates it from the metacone. The slightly raised lingual rim of this groove is probably a vestigial metaconule. A tiny, bump-like, vestigial paraconule is present on the anterior edge of the talon, between the protocone and the base of the paracone. There is a strong size disparity between the paracone and the metacone. In lingual view the paracone is a mere bulge on the side of the tall, pyramidal metacone. Extensive wear has removed the apex and the anterior side of the cusp, obliterating the paracristae. The base of the premetacrista forms a slightly taller, rounded blade, indicating that a carnassial notch was originally present between the postparacrista and the premetacrista.The tall conical metacone dominates the crown. Only a small wear facet is developed at its tip. The premetacrista extends steeply down the anterior face of the metacone, parallel with the anterior–posterior axis of the tooth, indicating that the centrocrista was probably straight when the postparacrista component was present. The metacone is a tall conical cusp that dominates the tooth. Only a small wear facet has developed at the tip of the cusp. The postmetacrista forms a sharp edge that extends down the posterior side of the metacone and then continues posterobuccally as a horizontal edge along the posterior margin of the metastylar wing. This part of the postmetacrista is raised slightly above the level of the buccal rim of the stylar shelf so that the metastylar basin is tilted slightly buccally unlike that of M2. The posterobuccal corner of the crown, where the postmetacrista meets the stylar crest is slightly raised producing a vestigial metastyle. The stylar crest is bowed between the metastyle and stylar cusp D in buccal view. Stylar cusp D is smaller and set lower than it is in M2. The ectoflexus interrupts the stylar crest with a low rounded saddle joining stylar cusp D with an anterior stylar crest that rises gently to the low summit of stylar cusp B. As in M2 there is a small pit developed lingual to the ectoflexus, adjacent to the base of the metacone.

Table 4 Measurements of upper molars of Thylacinus potens and Th. cynocephalus.

Data for CPC 6746 and Th. cynocephalus are taken from Woodburne (1967). Measurements for Th. cynocephalus are mean values taken from a sample of six specimens. Measurements are in mm.

	M1W1	M2L	M2W1	M2W2	M3L	M3W1	M3W2	M4L	M4W1	M4W2	
Th. potens											
NTM P4326	12.4	15.7	14.7	18.0	16.0	15.5	18.6	11.5	14.0	9.5	
NTM P4379	–	16.1	∼14.1	∼17.7	–	–	–	–	–	–	
CPC 6746	12.8	15.7	13.9	17.5	15.2	15.9	19.0	12.2	15.8	9.9	
Th. cynocephalus											
Mean	7.8	13.2	10.0	15.0	15.1	12.0	17.8	9.7	12.6	7.8	
Notes.

L mesiodistal length

W1 width of the crown from the mesiobuccal corner to the lingual side of the protocone

W2 width of the crown from the metastylar corner to the lingual side of the protocone

M4 is complete. As in other dasyuromorphians it is reduced in size relative to the preceding molars and is strongly oriented posterolingually. The metastylar wing is strongly reduced in comparison to those of the preceding molars and there is only one large cusp, the paracone, occupying the central region of the tooth, buccal to the protocone. A short but distinct parastylar crest occupies the anterobuccal corner of the tooth. The posterobuccal margin is evenly concave between the metastyle and the parastylar crest, rather than possessing the distinct ectoflexus seen in the preceding molars. The posterobuccal face of the crown slopes strongly down to this margin from the apex of the paracone and is not offset by a stylar shelf. The anterior margin between the parastylar crest and the paracone is distinctly convex in occlusal view. The steeply sloping buccolingual face curves outward at the base of the crown in this region to form a weakly developed precingulum. Weak inflections in occlusal view separate the reduced, U-shaped protocone from the rest of the tooth. The protocone forms a small pointed tubercle that is set lower than the rest of the tooth. The lingual side of the cusp curves buccally toward the tip so that it is set away from the lingual margin and close to the groove separating the protocone from the lingual base of the paracone. The very short pre- and post- protocristae extend close to vertically down the anterior and posterior edges of the buccal face of the protocone. The paracone is the largest cusp of the tooth and forms a central, pyramidal projection. A near vertically oriented wear facet occupies the anterior side of the tooth between the paracone and the parastylar crest, above the precingulum. Two elongate cristae extend from the paracone. The longest of these is the preparacrista which extends in a straight line buccoanteriorly to the parastylar crest. In posterior view the crista slopes gently down from the paracone. The shorter postparacrista extends posteriorly to the metastyle. In lingual view this crista slopes downward at a steep angle. The short parastylar crest developed at the buccoanterior corner of the tooth bears two minute cuspules which are presumably the parastyle and stylar cusp B.

Dentary

The dentary specimen (NTM P4327) contains the canine alveolus, roots of P1, damaged and incomplete P2, P3, M1 and M2 and complete M3 and M4 (Figs. 12, 13 and 14). The anterior tip of the dentary is crushed and the tip carrying the incisors and their alveoli is missing. Posteriorly the dentary has broken off at the level of the anterior rise of the coronoind process. The dentary is relatively slender and transversely compressed, although the latter may have been accentuated by post-mortem compaction. The dentary depth below the posterior root of M4 is 33.2 mm, which lies within the range of Th. cynocephalus (Table 2). The medial symphyseal surface extends posteriorly to a level near the posterior end of P3. In lateral view the anterior tip is acutely pointed and the ventral margin in forms a gentle convex curve along its entire length. The ventral margin between P2 and M1 is expanded laterally forming a low ventrolateral torus (Fig. 13A). The lateral surface is depressed above the thickened ventral margin and bears three mental foramina below P2, P3 and M2 respectively. An anterior mental foramen may be present below P1 but crushing and fragmentation of the dentary surface in this area prevents accurate determination. Posteriorly the lateral surface of the dentary is excavated by the masseteric fossa. The ventral and anterior margins of the fossa are indistinct but it is bordered anterodorsally by a ridge that continues posterodorsally to form the leading edge of the coronoid process.

Figure 11 Heavily worn upper molar.

Thylacinus potens. NTM P4379, heavily worn right M2 in a fragment of the right maxilla. (A) photograph of occlusal view, (B) photograph of buccal view, (C) photograph of lingual view, (D) interpretive drawing of A, (E) interpretive drawing of B, (F) interpretive drawing of C. Abbreviations: abr, anterior buccal root; ef, ectoflexus; lr, lingual root; ms, metastyle; mx, maxilla fragments; pbr, posterior buccal root; pc, precingulum; pr, protocone; sB, stylar cusp B. Grey fill represent areas of adherent matrix, areas hatched with continuous horizontal lines represent broken tooth surfaces, areas hatched with discontinuous lines represent wear surfaces. Scale bar = 10 mm.

Figure 12 Photographs of dentary

Thylacinus potens. NTM P4327, photographs of horizontal ramus of left dentary. (A) lateral view, (B) occlusal view, (C) medial view. Scale bar = 50 mm.

Figure 13 Drawings of dentary.

Thylacinus potens. NTM P4327, interpretive drawings of Fig. 11. (A) lateral view, (B) occlusal view, (C) medial view. Abbreviations: arp1, anterior root of first premolar; ca, lower canine alveolus; df, digastric fossa; m1–4, molars 1–4; mf, masseteric fossa; p1–3, premolars 1–3; pmf, posterior mental foramina; prp1, posterior root of first premolar; sym, symphyseal surface; vt, ventrolateral torus. Grey fill represent areas of adherent matrix, areas hatched with continuous horizontal lines represent broken bone and tooth surfaces, areas hatched with discontinuous lines represent wear surfaces. Scale bar = 50 mm.

Dentary dentition

Although the incisor-bearing area is missing there is very little space between the anterior projection of the symphyseal surface and the broken anteromedial margin of the jaw tip, indicating that the incisors must have been small and crowded. The large canine was placed close to the anterior tip of the dentary and apparently projected anterodorsally. An isolated lower canine (NTM P4461) does not differ from those of Th. cynocephalus (Fig. 16). The cheek teeth were closely spaced with all teeth contacting their adjacent teeth except for a short diastema of 4.6 mm between P1 and P2. As in the upper tooth row of the holotype specimen the long axis of P1 is obliquely oriented in relation to P2 and P3 (Figs. 12B and 13B).

Figure 14 Closeups of lower molars in occlusal view.

Thylacinus potens. NTM P4327, detail of lower molar tooth row in occlusal view. (A) posterior molars, (B) first molar, (C) interpretive drawing of A, (D) interpretive drawing of B. Abbreviations: cdo, cristid obliqua; cn, carnassial notch; ed, entoconid; hd, hypoconid; pad, paraconid; pcd, postcristid; pcid, precingulid; ppcd, postparacristid; prd, protoconid; prpcd, preprotocristid. Grey fill represent areas of adherent matrix, areas hatched with continuous horizontal lines represent broken tooth surfaces, areas that are hatched with discontinuous lines represent wear surfaces. Scale bar = 20 mm.

Figure 15 Closeups of lower molars in buccal view.

Thylacinus potens. NTM P4327, detail of lower molar tooth row in buccal view. Abbreviations: cdo, cristid obliqua; cn, carnassial notch; ed, entoconid; hd, hypoconid; pad, paraconid; pcd, postcristid; pcid, precingulid; prd, protoconid; prpcd, preprotocristid. Grey fill represent areas of adherent matrix, areas hatched with continuous horizontal lines represent broken tooth surfaces, areas that are hatched with discontinuous lines represent wear surfaces. Scale bar = 20 mm.

Figure 16 Lower canine.

Thylacinus potens. NTM P4461, right lower canine. (A) buccal view, (B) posterior view. Scale bar = 10 mm.

Only the posterior part of P2 is preserved. It indicates a tall, buccolingually compressed triangular tooth, with a rounded posterior margin that descends to the base of the crown without any expansion to form a posterior heel-like cuspid.

The apex of the central protoconid is missing from P3. Nonetheless it is clear that it was similar to P2 in both size and shape (Table 5). It differs in being buccolingually thicker, and having a concave posterior margin in lateral view that forms a weakly-developed heel-like posterior cuspid. A slight bulge on the anterior profile of the tooth indicates an incipient paraconid.

Table 5 Measurements of lower premolars of Thylacinus potens and Th. cynocephalus.

Measurements for Th. cynocephalus are mean values taken from a sample of six specimens. Measurements are in mm.

	P1W	P1L	P2W	P2L	P3W	P3L	
Th. potens							
NTM P4327	∼4.9	12.7	5.6	15.5	6.8	14.8	
Th. cynocephalus							
Mean	3.4	6.0	4.1	9.1	5.0	10.6	
Notes.

L mesiodistal length

W maximum buccolingual width

The central protoconid of M1 is heavily worn but the tooth clearly displays a low rounded paraconid with a worn tip, anterior the base of the protoconid. A weak notch on the lingual side of the tooth separates the two cusps. The buccal surface of the bases of these two crowns forms a continuous surface that faces slightly anteriorly and apically. The anterobuccal margin is slightly thickened to form a vague hint of a cingulid. Posterior to the paraconid is a anteroposteriorly short and buccolingually broad talonid shelf. The talonid is slightly wider than the trigonid (Table 6). The posterior and lingual sides of the shelf are close to vertical while the buccal side forms an apicolingually sloping surface. The talonid shelf bears a flattened wear surface on its buccal side that represents a worn hypoconid. A shallow anteroposteriorly oriented groove separates this worn area from a low rounded entoconid developed on the lingual side of the talonid.

Table 6 Measurements of lower molars of Thylacinus potens and Th. cynocephalus.

Data for UCMP 66206 are taken from Woodburne (1967). Data for Th. cynocephalus are mean values taken from Woodburne (1967) and Dawson (1982). Measurements are in mm.

	M1L	M1W1	M1W2	M2L	M2W1	M2W2	M3L	M3W1	M3W2	M4L	M4W1	M4W2	
Th. potens													
NTM P4326	14.8	6.7	7.9	16.3	(7.3)	8.5	15.3	8.6	8.5	17.7	9.9	6.5	
UCMP 66206	–	–	–	13	6.8	6.8	14.5	8.3	6.2	15.4	8.8	5.1	
Th. cynocephalus													
Mean (Woodburne)				12.0	5.7	6.2	14.0	6.9	6.8	16.0	7.8	4.3	
Mean (Dawson)	9.6	4.4		12.0	5.7		14.1	6.9		15.7	7.6		
Notes.

L mesiodistal length

W1 width of the trigonid

W2 width of the talonid

Most of the crown of M2 is missing with the edges worn and rounded suggesting that this tooth was lost during the life of the animal. The worn talonid is slightly broader buccolinually than the talonid of M1. No other details of this tooth are apparent.

M3 is well preserved although the linguoanterior corner of the tooth is missing, preventing determination of the height of the paraconid. A narrow but well-developed buccoanterior cingulid slopes steeply posteroventrally from the anterior base of the paraconid to the base of the crown at the level of the anterior margin of the protoconid. The protoconid forms a tall, narrowly triangular cusp in lateral view. The tip is worn with an anterodorsally facing facet intersects the posterior wear facet producing a short transversely aligned crest at the tip of the protoconid. A weakly developed preprotocristid extends down the anterior surface of the protoconid to terminate at the base of the paraconid, on its buccal side. A near vertical, posterobuccally facing wear facet occupies the posterior surface of the protoconid. The postprotocristid forms a slightly raised carina along the lingual margin of this wear facet. The postprotocristid extends from the tip of the protoconid and terminates in the notch between the hypoconid and the protoconid. The talonid is a low, anteroposteriorly short shelf that is slightly wider than the trigonid. It bears two main cuspids: the hypoconid and hypoconulid, with a vestigial trace of the entoconid. The hypoconid is worn flat and its roughly circular base is set lingually from the buccal margin, resulting in a sloping buccal side of the talonid. The basal wear facet of the hypoconid lies abuts the base of the protoconid, with just a narrow notch separating them. Thus the cristid obliqua, which would have formed one half of a carnassial notch, has been obliterated. A postcristid extends a short distance from the lingual side of the hypoconid, along the posterior margin of the talonid shelf to the low, pyramidal hypoconulid. Immediately lingual to the hypoconulid, at the posterolingual corner of the talonid, is a bump-like vestige of the entoconid. The lingual side of the talonid is open and the floor of the talonid basin curves downward onto the lingual side of the shelf here.

M4 is complete and well preserved. Its anterior–posterior length is greater than that of M3 (Table 6). A well-developed, conical paraconid forms the second highest cusp of the tooth. It arises from the linguoanterior corner of the tooth. The buccoanterior surface is coplanar with the anterobuccal surface of the protoconid and is bordered basally by a buccoanterior cingulid, similar to that seen in M3. A sharply incised groove separates the posterobuccal surface of the paraconid from the protoconid. The large protoconid is a tall, conical cusp that forms the highest point of the tooth. In buccal view it’s relatively taller than in Th. cynocephalus, with a straight as opposed to gently convex anterior margin. The groove separating the protoconid from the paraconid is narrower and far shallower than the prominent carnassial notch present in Th. cynocephalus. A weak preprotocristid extends from the buccal end of this groove to the apex of the protoconid. As in M3 there is a nearly vertical wear facet developed on the posterior surface of the protoconid. A distinct postprotocristid forms the lingual border of this wear facet. It extends steeply down the posterior face of the protoconid and meets the postcristid described below in a small carnassial notch. The area occupied by the talonid is reduced relative to the preceding molars. A single cuspid, apparently the hypoconulid, arises from the posterolingual corner of the talonid. This forms a moderately tall conical process that stands 4.5 mm above the posterior base of the crown. A short postcristid curves anterobuccally from the hypoconulid to join the posterior base of the protoconid. A small swelling on this cristid, where it meets the base of the protocone may represent a reduced remnant of the hypoconid. The lingual side of the talonid shelf is not bordered by any cristid or cuspid and the floor of the shelf slopes downward at its lingual side.

Discussion

Autapomorphies of Th. potens

Woodburne (1967) provided an extensive list of characters that distinguished Th. potens from Th. cynocephalus. Since that time an extensive range of pre-Pleistocene thylacinids have been discovered. These indicate that many of the diagnostic characters proposed by Woodburne are widely distributed among pre-Pleistocene thylacinids and represent plesiomorphic characters that are general for thylacinids. Other proposed diagnostic characters can now be shown to vary within Th. potens, with the addition of new specimens described above. The following five characters stand as unambiguous autapomorphies of Th. potens:

Long axis of P1 mesiobuccally oriented in adults (modified from Murray, 1997). In occlusal view the anterior–posterior axis of the first upper premolar is aligned with the canine and the subsequent premolars in most adult thylacinids including Th. cynocephalus (SAM M95, M1959), Th. megiriani (NTM P9618), Nimbacinus dicksoni (Wroe & Musser, 2001, Figs. 1B, 4) and Badjcinus turnbulli (Muirhead & Wroe, 1998, Fig. 2B). In juvenile Th. cynocephalus the anterior end of P1 is rotated buccally so that the anterior–posterior axis is canted anterobuccally (SAM M1956). This is probably related to tooth crowding in juveniles since adult specimens of Th. cynocephalus have normally aligned first upper premolars. In contrast adult specimens of Th. potens have the out-turned condition.

Anterior width of the first upper molar greater than its anterior–posterior length. Primitively the anterior–posterior length of M1 exceeds the anterior width (the width from the protocone to the mesiobuccal corner of the tooth) in thylacinids. This condition is present in dasyurids (e.g., Dasyurus maculatus: NTM U7542; Antechinus flavipes: NTM U7566) and all known species of thylacinids (e.g., N. richi: NTM P9973-11; Th. cynocephalus: Woodburne, 1967, Table 1) except Th. potens (Woodburne, 1967, Table 1) where the anterior width exceeds the anterior–posterior length.

Reduced palatal fenestrae. As described by Woodburne (1967) the palatal fenestrae of Th. potens lie below the range of dimensions displayed by Th. cynocephalus, despite coming from a larger palate. Relative to the upper molar row length, the length of a palatal fenestra is about 33% in Th. potens whereas this proportion ranges between 50 and 58% in adult Th. cynocephalus (based on the author’s observation of SAM specimens). All of the few preserved palates of other older thylacinids have relatively large palatal fenestrae like those of Th. cynocephalus (e.g., 55% in Mutpuracinus archibaldi: NTM P91168-5; 53% in N. dicksoni: Wroe & Musser, 2001, Fig. 1B; and a fenestra that “extends from M1 to M3” in B. turnbulli: Muirhead & Wroe, 1998, p. 613). Thus the reduced condition seen in Th. potens is, as far as can be determined, an autapomorphy of the species.

Absence of a diastema between P1 and P2. Most thylacinid species possess a diastema between the first and second lower premolars, e.g., B. turnbulli (Muirhead & Wroe, 1998, Fig. 1A), N. dicksoni (Wroe & Musser, 2001, Fig. 2B), Mut.archibaldi (Murray & Megirian, 2006), Th. macknessi (Muirhead & Gillespie, 1995, Fig. 1A) and Th. cynocephalus (SAM M1959). The sole known exceptions are N. richi (NTM P9612-4) and Th. potens (NTM P4327). Given that N. richi is phylogenetically remote from Th. potens, this character can be interpreted as an unambiguous autapomorphy of Th. potens that has convergently evolved in N. richi.

Relative enlargement of P2 so that it is longer than P3 and M1. Primitively the longest lower premolar of thylacinids is the posterior one, here designated P3. This condition is present in Muribacinus gadiyuli (Wroe, 1996, Fig. 1.4), B. turnbulli (Muirhead & Wroe, 1998, Table 2), Mut. archibaldi (Murray & Megirian, 2006, Table 2), N. dicksoni (Wroe & Musser, 2001, Fig. 2), N. richi (Murray & Megirian, 2000, Table 1), Wabulacinus ridei (Muirhead, 1997), Th. macknessi (Muirhead & Gillespie, 1995, Table 1), Th. megiriani (NTM P4376) and Th. cynocephalus (SAM M95, M1959). Th. potens is unique in having P2 exceed P3 in length (Table 5). Not only does P2 exceed P3 but it also exceeds M1 (Table 6), indicating that it is P2 that has undergone relative enlargement.

In addition to these characters a further pair of characters are ambiguous autapomorphies of Th. potens that due to their shared presence in other taxa phylogenetically close to Th. potens can be equally interpreted as ambiguous autapomorphies of Th. potens or transient synapomorphies of more inclusive clades.

Ventrally facing sulcus ventral to the maxillary root of the zygomatic arch. Both the holotype and referred maxillae of Th. potens possess a distinct, ventrally facing sulcus that incised along the ventral margin of the root of the zygomatic arch. In NTM P4326 this sulcus starts just above the midlength of M3 and continues posterodorsally onto the base of the zygomatic arch, posterior to M4. Although the posterior end of the maxilla is missing in the holotype and only known maxilla of Th. megiriani, the anterior end of a similar sulcus can be seen dorsal to the empty alveolus for M4. No other thylacinids or dasyurids appear to have a comparable sulcus. It can therefore be interpreted as a convergence between Th. potens and Th. megiriani, or a transient synapomorphy of large-bodied Thylacinus species that was reversed in Th. cynocephalus.

P2 longer than M1. Primitively the second and third upper premolars of thylacinids have shorter crowns than the first upper molar. In Ty. rothi, Th. potens and Th. megiriani P3 is enlarged so that it is longer than the first molar. This appears to be a synapomorphy of derived thylacinids including Tyarrpecinus and Thylacinus that is reversed in Th. cynocephalus. However it is only in Ty. rothi and Th. potens that both P2 and P3 are longer than M1. This is either an autapomorphy of Th. potens that is convergently developed in Ty. rothi, or it is a transient synapomorphy that is reversed in Th. megiriani and Th. cynocephalus. The presently unknown anterior upper dentitions of W. ridei and Th. macknessi would decide which one of these alternatives is the more parsimonious.

Variation within Th. potens

The new specimens display some distinctive differences from the original hypodigm described by Woodburne (1967). The new maxilla differs from the holotype in a number of respects, some of which resemble the modern Th. cynocephalus. The holotype of Th. potens displays a wide (4 mm) diastema between the canine and P1, whereas no such diastema is present in NTM P4326. Murray (1997) noted the anterior position of the infraorbital foramen above the posterior end of M1 in the holotype specimen of Th. potens and suggested it may be related to facial shortening. However the infraorbital foramen opens above M2 in NTM P4326 as it does in Th. cynocephalus.

Woodburne (1967) also noted that the anterior palate of the holotype was longitudinally bowed, with a broad low ridge separating the anterior depressed area bearing the incisive foramina from the rest of the palate. The anterior palate of NTM P4326 is however simple and flat, like that of Th. cynocephalus. Dentally the new maxilla also displays a few differences from that of the holotype, namely the ectoflexus of M2 is more weakly developed than that of M3, although it is still more prominent than in the M2 of Th. cynocephalus. Lastly Woodburne (1967) noted that M2 and M3 were subequal in size in the holotype, with M3 being slightly shorter than M2. In NTM P4326 M3 exceeds M2 in both length and width (Table 4), although the discrepancy between the two molars is not as great as that displayed by Th. cynocephalus.

Even more dramatic differences can be seen between Woodburne’s dentary and the new dentary. As Woodburne (1967) observed, the paratype dentary fragment of Th. potens has an unusually deep dentary below the posterior molars, both in absolute measurements and relative to the length of the posterior molars. The depth of UCMP 66206 below M4 is 37.0 mm, or 2.40 times the length of M4. In contrast the same measurement is 31.2 mm, or 1.71 times the length of M4, in NTM P4327 which lies within the range displayed by a sample of recent Th. cynocephalus (Table 2). Further differences between these specimens can be seen in the teeth. Firstly, there is a well-defined precingulid on M3–4 of NTM P4327, whereas Woodburne (1967, pg. 35) indicated that there is “only a faint suggestion of a cingulum” in this position on the M3 of UCMP 66206. Secondly the talonid of M3 is not transversely reduced in NTM P4327 whereas it is distinctly narrower than the trigonid in UCMP 66206. The latter character is another feature that NTM P4327 shares with Th. cynocephalus.

On the basis of these comparisons it would be prudent to question whether the new specimens truly belong to Th. potens, or in fact represent a taxon that is more closely related to Th. cynocephalus. The latter hypothesis is considered less likely than the former for two reasons. Firstly the new specimens share apomorphic character states with Th. potens that are not seen in Th. cynocephalus. These include: molar row lengths exceeding those of Th. cynocephalus (both NTM P4326 and P4327, see below); anterior end of P1 out-turned (NTM P4326); presence of a sulcus on the ventral margin of the maxillary root of the zygomatic arch (NTM P4326); P2 longer than M1 (NTM P4326). Although the type series of Th. potens lacked an anterior end of a dentary, the presence of an enlarged P2 in NTM P4327 would match the derived condition of an enlarged second premolar seen in the upper dentition. In contrast, the similarities shared between these new specimens and Th. cynocephalus are symplesiomorphies that are general to thylacinids. Thus whatever NTM P4326 and P4327 are, their relationship appears to be closer to Th. potens than to any other known thylacinid. Secondly if one were to treat the new specimens as representing a second taxon then this would imply two, closely-related, large-bodied apex predators living as contemporaries in the same local fauna. This is a most unlikely situation.

Thus it would appear that Th. potens is a variable species. While much of this variation does not appear unusual in comparison to the variation displayed by historic specimens of Th. cynocephalus or other extant dasyuromorph species, the variation seen in the ratio of dentary depth to M4 length is unusually large in Th. potens. This ratio ranges from 1.49 to 1.76 in a historic sample of nine adult Th. cynocephalus, compared to the range of 1.8–2.4 seen in just two Th. potens specimens. Given that both the new dentary and Woodburne’s specimen show complete eruption of all teeth and an advanced stage of tooth wear it is apparent that they represent mature individuals. Therefore ontogenetic differences cannot account for the differences in relative mandibular depth. A larger sample size is required to test the possibility that the observed variation is the result of sexual dimorphism. Sexual dimorphism is known to have existed in Th. cynocephalus with males having linear skull dimensions 13–86% larger than those of females (Jones, 1997).

Phylogenetic position of Th. potens

The discovery of the anterior end of the dentary of Th. potens revealed an unexpected plesiomorphic character state. All three premolars are set adjacent to one another, with no diastemata between them or the following M1. Diastemata occur between P2 and P3 of all other species of Thylacinus but are absent from more basal thylacinids such as W. ridei (Muirhead, 1997), Mut. archibaldi (Murray & Megirian, 2006) and B. turnbulli (Muirhead & Wroe, 1998). A number of other previously known plesiomorphies distinguish Th. potens from other Thylacinus species, these include the retention of a precingulum on M3, M3 that is slightly wider than it is long, and M2 and M3 that are subequal in length. These character states were included in a cladistic analysis to test whether the additional data was enough to cause Th. potens to fall outside the genus Thylacinus.

The analysis returned 2 most-parsimonious trees of 88 steps. The strict consensus of the two trees is well-resolved (Fig. 17A). The only polytomy, encompasses the base of Thylacinus and the two taxa found to be most closely related to this genus, i.e., W. ridei and Ma. muirheadae. Inspection of the trees reveals that is the position of Ma. muirheadae that varies between the two. This is unsurprising given that this species is known from just a single tooth, making it the most poorly-known thylacinid and only scorable for 21% of the characters used in this analysis. If Ma. muirheadae is pruned a posteriori from the most parsimonious trees, a single, fully-resolved, reduced consensus tree is obtained.

Figure 17 Results of cladistic analysis of thylacinid relationships.

Consensus trees of two most-parsimonious-trees (tree length = 88 steps) resulting from a cladistic analysis of 13 thylacinid taxa with Dasyuridae set as the user-defined outgroup. (A) strict consensus with bootstrap support values for clades with support values >50%, (B) fully resolved reduced cladistic consensus obtained after a posteriori pruning of Maximucinus muirheadae. Letters at nodes correspond to those in the tree description in Appendix 2.

The two most parsimonious trees both resolve Th. potens as the sister taxon to the other large-bodied late Neogene species (Th. cynocephalus and Th. megiriani) within the genus Thylacinus (Fig. 17), supporting all previous assessments of the relationships of this species. The clade uniting the three large Thylacinus species has a moderate level of bootstrap support (69%) but this value is lowered by the instability of Ma. muirheadae. When a second bootstrap analysis is conducted, with Ma. muirheadae excluded, the bootstrap support for this clade jumps to 83%, indicating it is a robust result. Thus the plesiomorphic characteristics of Th. potens that are listed above are interpreted as character reversals.

Size of Thylacinus potens

While several authors have noted the greater robustness and likely greater size of Th. potens relative to the modern Th. cynocephalus (e.g., Woodburne, 1967; only Wroe, 2001) has attempted a quantatitive estimate of the body mass of Th. potens. He found the holotype specimen to have come from an individual weighing 38.7 kg but noted that the estimate was based on the combined length of M1–3 and an assumption of geometric similitude to Th. cynocephalus. The results of the size estimates for the new specimens are summarised in Table 7.

Table 7 Mass estimates for Thylacinus potens.

Mass estimates for two of the new specimens of Thylacinus potens. Regression equations derived by Myers (2001) from his dasyuromorphian dataset.

Specimen	Method	Regression equation	Measurement
(mm)	Smearing
estimate (%)	Mass
estimate (kg)	
NTM P4326	Regression of 2UMW	log y = 0.379 + 4.038(log x)	14.7	3	120.6	
	Regression of estimated UMRL	log y = −0.992 + 3.279(log x)	51	1.2	40.9	
	Geometric similitude		43.2		43.3	
NTM P4327	Regression of LMRL	log y = −1.075 + 3.209(log x)	63.3	3	56.1	
	Geometric similitude		63.3		56.0	
Notes.

2UMW width of the second upper molar

UMRL upper molar row length

LMRL lower molar row length

As can be seen, the different estimates for each specimen are remarkably close to one another with the exception of the estimate based upon regression of the width of M2. The estimate of 121 kg is clearly far too high and indicates that Th. potens had relatively broader second upper molars in comparison to other dasyuromorphians.

It is also interesting to note that all of the estimates exceed the value of 38.7 kg that Wroe (2001) obtained for the holotype of Th. potens by inferring geometric similitude with Th. cynocephalus. While the estimates based on the regressions of Myers come with the caveat that they extrapolate beyond the sample used to generate the regression, they do support the hypothesis that Th. potens attained a larger size than the modern thylacine which had an average body weight of 29.5 kg (Paddle, 2000) and a maximum reconstructed weight of 35 kg (Moeller, 1968).

However there is the additional caveat that these estimates assume that Th. potens had not evolved unusual body proportions that strongly departed from geometric similitude with Th. cynocephalus or the scaling of other dasyuromorphians. Relatively few postcranial elements for Th. potens are known and are still under study by the author. However, an adult humerus was recovered in Shattered Dreams close to NTM P4326 and NTM P4327 and is smaller than average for Th. cynocephalus and hints that the proportions of Th. potens may have indeed been unusual. Further study of other postcranial remains is required to determine if this humerus is typical of Th. potens or from an unusually small individual.

Palaeobiology

The sample of Th. potens teeth display heavy damage and wear including the wearing down of P2 to a rounded stump, the strongly blunted protocone of P3, the virtual obliteration of the paracones from M2 and M3 in NTM P4326, the wearing down of M2 to a single flat plane in NTM P4379 (Fig. 11), the virtual obliteration of the protocone by wear in NTM P4516 and the almost complete loss of the crown of M2 in NTM P4327 (Figs. 12 and 13). In comparison, a sample of eight adult skulls of Th. cynocephalus (SAM M95, M922, M1953, M1954, M1955, M1959, M1960 and M665/001) show no premolars worn to rounded stumps, no molars that have been worn to a flat plane, no paracones or protocones obliterated by wear, only one instance of a lower molar cusp (M4 paraconid) missing due to breakage and just one instance of a cheek tooth missing due to pre-mortem breakage. These observations strongly suggest that Th. potens were much harder on their cheek teeth than Th. cynocephalus and may have been indulging in durophagy, quite possibly bone-cracking, a feeding style that Th. cynocephalus was ill-equipped to perform (Attard et al., 2011). It is somewhat puzzling then that the teeth of Th. potens do not show a strong trend towards bone-cracking adaptations. For example bone-cracking mammals tend to develop the following features: well developed cingula and cingulids; broad, low crowned premolars and lower molars, lower broader molar cusps and a migration of the molar cusps toward the centre of the tooth crowns (Wroe, 1998). Th. potens shows no trend towards these features over the character states present in thylacinids basally. Thus it is possible that frequent bone cracking was a relatively new behaviour in Th. potens and that morphological specialisations had yet been given sufficient time to evolve. Alternatively the few preserved individuals known for this species may have been exhibiting exceptional behaviour, possibly related to the unusual environmental conditions associated with the Alcoota mass death assemblage.The question can only be explored with the collection of a larger sample of specimens, more complete specimens or, ideally, the location of Th. potens specimens in a different depositional setting.

The new maxilla and dentary which form the basis of this paper were discovered in a new pit (‘Shattered Dreams’) which was opened thanks to the generous loan of a backhoe and licensed operator from Central Desert Shire, Northern Territory. I am deeply indebted to Glenn Marshall for making this loan possible. The specimens themselves were found and patiently excavated by Jared Archibald. I also wish to thank Catherine Kemper and Ben McHenry of the South Australian Museum for allowing me access to thylacinid specimens in their care. Karen Black and Gavin Prideaux provided thoughtfull reviews which improved the quality of the final paper. All photographs used in this paper were taken by Steven Jackson.

APPENDIX 1. Character List

1. Relationship of jugal to infraorbital foramen: jugal widely separated from the margin of the infraorbital foramen (0); maxilla-jugal suture passes very close to the margin of the infraorbital foramen (1); jugal contributes to the posterior margin of the infraorbital foramen (2). Modified from character 3 in Muirhead & Wroe (1998). Character is treated as ordered.

2. Position of the infraorbital foramen: infraorbital foramen is dorsal to M1 (0); infraorbital foramen is dorsal to M2 (1). Character 12 in Murray (1997).

3. Presence or absence of a sulcus along the ventral margin of the maxillary zygomatic root: sulcus absent (0); sulcus present (1). Character is new.

4. Enclosure of the primary foramen ovale: foramen ovale partly bordered by the periotic (0); foramen ovale completely enclosed by the alisphenoid, excluding periotic from its margin (1). Character 5 in Muirhead & Wroe (1998).

5. Relationship of the alisphenoid and petrosal tympanic processes: petrosal tympanic process contacts the alisphenoid tympanic process (0); petrosal tympanic process reduced or absent, so that it does not contact the alisphenoid tympanic process (1). Character 9 in Muirhead & Wroe (1998).

6. Position of P2: P2 closer to P3 than to P1; P2 equidistant between P3 and P1. Character is new.

7. Anterior–posterior length of P2: P2 shorter than M1; P2 longer than M1. Character is new.

8. Anterior–posterior length of P3: P3 shorter than M1; P3 longer than M1. Character is new.

9. Presence or absence of a posterolingual cuspule on P3: cuspule absent (0); cuspule present (1). Character 7 in Wroe & Musser (2001).

10. Development of precingulum on M1: precingulum present and complete extending from the anterobuccal corner to a point anterior to the base of the protocone (0); precingulum present but incomplete, extending from anterobuccal corner to a point anterior to the base of the paracone (1); precingulum absent (2). Character 12 in Muirhead & Wroe (1998). Character is treated as ordered.

11. Presence or absence of a precingulum on M3: precingulum present (0); precingulum absent (1). Character is new.

12. Orientation of the preparacrista on M1: preparacrista perpendicular to the long axis of M1 (0); preparacrista angled anterobuccally relative to the long axis of M1 (1). Character 16 in Muirhead & Wroe (1998).

13. Presence or absence of a postcingulum on M1: postcingulum present (0); postcingulum absent (1). Character 25 in Wroe & Musser (2001).

14. Development of the ectoflexus on M2 and M3: ectoflexus well-developed (0); ectoflexus extremely reduced or absent (1). Modified from Fig. 11 in Murray (1997).

15. Size of the paracone in the upper molars: large, approaching the size of the metacone (0); significantly reduced, much less than the size of the metacone (1). Modified from character 10 in Muirhead & Wroe (1998).

16. Shape of the centrocrista of M1 in occlusal view: sharply angled (0); obtusely angled (1); straight (2). Modified from character 15 in Muirhead & Wroe (1998). Character is treated as ordered.

17. Shape of the centrocrista of M2 and M3 in occlusal view: sharply angled (0); obtusely angled (1); straight (2). Modified from character 15 in Muirhead & Wroe (1998). Character is treated as ordered.

18. Elongation of the postmetacrista in upper molars: postmetacrista not elongate with the metastylar wing occupying 40–48% of the tooth length (0); postmetacrista mildly elongated with the metastylar wing occupying 48–52% of the tooth length (1); postmetacrista strongly elongated with the metastylar wing extending over 52% of the length of the tooth (2). Modified from character 14 in Muirhead & Wroe (1998). Character is treated as ordered.

19. Presence or absence of the protoconule on the upper molars: protoconule present (0); protoconule absent (1). Character 14 in Wroe & Musser (2001).

20. Presence or absence of the metaconule on the upper molars: metaconule present (0); metaconule absent (1). Character 15 in Wroe & Musser (2001).

21. Size of M3 relative to M2: M3 and M2 are subequal (0); M3 is distinctly larger than M2 (1). Character is new.

22. Shape of M3: M3 is as wide as, or wider than it is long (0): M3 is longer than it is wide (1). Character is new.

23. Size of stylar cusp B on M1 and M2: stylar cusp B is well-developed (0); stylar cusp B is highly reduced or absent (1). Modified from character 11 in Muirhead & Wroe (1998).

24. Presence or absence of stylar cusp C on M1: stylar cusp C is absent (0); stylar cusp C is present (1). Modified from character 21 in Wroe & Musser (2001).

25. Presence or absence of stylar cusp C on M2 and M3: stylar cusp C is present (0); stylar cusp C is absent (1). Modified from character 21 in Wroe & Musser (2001).

26. Development of stylar cusp D on M2: stylar cusp D is present and large (0); stylar cusp D is reduced to a slight bulge or a bump (1); stylar cusp D is absent (2). Character is treated as ordered.

27. Presence or absence of stylar cusp D on M3: stylar cusp D is present (0); stylar cusp D is absent (1). Modified from character 18 in Wroe & Musser (2001).

28. Presence or absence of stylar crest on M3: stylar crest present (0); stylar crest absent (1). Modified from character 7 in Muirhead (1997).

29. Presence or absence of a diastema between p1 and p2: diastema present (0); diastema absent (1). Character 22 in Muirhead & Wroe (1998).

30. Presence or absence of a diastema between p2 and p3: diastema absent (0); diastema present (1). Character 23 in Muirhead & Wroe (1998)

31. Anterior–posterior length of p3 relative to p2: p3 shorter than p2 (0); p3 longer than p2 (1). Modified from character 30 in Muirhead & Wroe (1998).

32. Presence or absence of a diastema between p3 and m1: diastema absent (0); diastema present (1). Character is new.

33. Development of metaconid in m1: metaconid is a well-developed, distinct cusp (0); metaconid is reduced to a small cuspule on the side of the protoconid (1); metaconid is absent (2). Modified from character 18 in Muirhead & Wroe (1998). Character is treated as ordered.

34. Development of metaconid in M2–4: metaconid is a distinct moderate-sized cusp (0); metaconid is reduced to a minute cuspule (1); metaconid is absent (2). Character 19 in Muirhead & Wroe (1998). Character is treated as ordered.

35. Position of the anterior termination of the cristid obliqua in lower molars: cristid obliqua terminates at the base of the protoconid (0); cristid obliqua extends partway up the posterior side of the protoconid (1); cristid obliqua extends to the tip of the protoconid (2). Character 29 in Muirhead & Wroe (1998). Character is treated as ordered.

36. Presence or absence of a carnassial notch in the cristid obliqua of lower molars: carnassial notch absent (0); carnassial notch present (1). Character 26 in Muirhead & Wroe (1998).

37. Presence or absence of a carnassial notch in the hypocristid of lower molars: carnassial notch absent (0); carnassial notch present (1). Character 26 in Muirhead & Wroe (1998).

38. Development of the entoconid in m1–3: entoconid a distinct, well-developed cusp (0); entoconid an indistinct cuspule or absent altogether (1). Modified from character 20 in Muirhead & Wroe (1998).

39. Anterior–posterior length of m4 relative to m3: m4 shorter than m3 (0); m4 longer than m3 (1). Character 32 in Muirhead & Wroe (1998).

40. Development of postcingulid on m1–3: postcingulid well-developed (0); postcingulid weakly developed (1). Character 36 in Wroe & Musser (2001).

41. Development of postcingulid on m4: postcingulid well-developed (0); postcingulid weakly developed (1). Character 37 in Wroe & Musser (2001).

42. Body size: dental measurements consistent with a body mass of less than 15 kg (0); dental measurements consistent with a body mass of 15–35 kg (1); dental measurements consistent with a body mass of greater than 35 kg (2). Modified from Fig. 11 in Murray (1997). Character is treated as ordered.

APPENDIX 2. Tree Description

The tree described here is the reduced strict cladistic consensus, which has had Maximucinus muirheadae pruned from it a posteriori. Designated letters for each clade correspond to those in Fig. 16B. Character state changes are given in brackets after each character number. Characters that have a CI of 1 (i.e., change only once and are free of homoplasy) are marked with an asterix.

Clade A. Thylacinidae

Content. Muribacinus gadiyuli, Badjcinus turnbulli, Ngamalacinus timmulvaneyi, Mutpuracinus archibaldi, Nimbacinus dicksoni, Nimbacinus richi, Tyarrpecinus rothi, Wabulacinus ridei, Thylacinus macknessi, Thylacinus potens, Thylacinus megiriani and Thylacinus cynocephalus.

Unambiguous synapomorphies. Character 9 (0 to 1): presence of a posterolingual cuspule on P3. Reversed at clade K (acctran) or in Th. cynocephalus (deltran). Ambiguous synapomorphy under acctran optimisation. Character 5 (0 to 1): Petrosal tympanic process strongly reduced so that it does not contact the alisphenoid tympanic process. Reversed in Mutpuracinus archibaldi.

Clade B

Content. Badjcinus turnbulli, Ngamalacinus timmulvaneyi, Mutpuracinus archibaldi, Nimbacinus dicksoni, Nimbacinus richi, Tyarrpecinus rothi, Wabulacinus ridei, Thylacinus macknessi, Thylacinus potens, Thylacinus megiriani and Thylacinus cynocephalus.

Unambiguous synapomorphies. Character 31 (0 to 1): P3 is longer than P2. Reversed in Thylacinus potens. Character 33 (0 to 1)*: Metaconid of M1 is reduced to a small cuspule. Character 35 (0 to 1)*: Anterior end of cristid obliqua extends partway up the posterior slope of the protoconid.

Ambiguous synapomorphy under deltran optimisation. Character 5 (0 to 1): Petrosal tympanic process strongly reduced so that it does not contact the alisphenoid tympanic process. Reversed in Mutpuracinus archibaldi.

Clade C

Content. Badjcinus turnbulli and Ngamalacinus timmulvaneyi.

Unambiguous synapomorphies. Character 37 (0 to 1)*: presence of a carnassial notch in the hypocristid. Character 40 (0 to 1): presence of a posterior cingulid on M4. Convergent in Th. macknessi.

Clade D

Content. Mutpuracinus archibaldi, Nimbacinus dicksoni, Nimbacinus richi, Tyarrpecinus rothi, Wabulacinus ridei, Thylacinus macknessi, Thylacinus potens, Thylacinus megiriani and Thylacinus cynocephalus.

Unambiguous synapomorphies. Character 4 (0 to 1)*: primary foramen ovale completely enclosed by the alisphenoid. Character 13 (0 to 1): loss of the postcingulum on M1. Reversed in Thylacinus megiriani.

Clade E

Content. Mutpuracinus archibaldi, Nimbacinus dicksoni, Nimbacinus richi.

Unambiguous synapomorphy. Character 24 (0 to 1): Presence of stylar cusp C on M1. Convergent in Barinya wangala and Thylacinus megiriani.

Clade F. Nimbacinus

Content. Nimbacinus dicksoni, Nimbacinus richi.

Unambiguous synapomporphy. Character 30 (0 to 1): Presence of a diastema between P2 and P3. Convergent in Muribacinus gadiyuli, and in Thylacinus with a reversal in Th. potens (acctran), or convergent in Muribacinus gadiyuli , Thylacinus macknessi and clade K (deltran).

Clade G

Content. Tyarrpecinus rothi, Wabulacinus ridei, Thylacinus macknessi, Thylacinus potens, Thylacinus megiriani and Thylacinus cynocephalus.

Unambiguous synapomorphies. Character 8 (0 to 1): P3 longer than M1. Convergent in Barinya wangala and reversed in Thylacinus cynocephalus. Character 16 (0 to 1)*: Wide, obtuse angle between the postparacrista and premetacrista of M1, creating a nearly straight centrocrista. Character 18 (0 to 1)*: mild elongation of the postmetacrista in M2 and M3 so that it is greater than 48% of the total length of the tooth. Character 38 (0 to 1)*: Loss of a distinct entoconid.

Ambiguous synapomorphies under acctran optimisation. Character 3 (0 to 1): Presence of sulcus along the ventral margin of maxillary zygomatic root. Reversed in Thylacinus cynocephalus. Character 7 (0 to 1): P2 longer than M1 Reversed in clade K. Character 10 (0 to 1): Reduction of the precingulum of M1 to an incomplete cingulum that does not reach the talon. Reversed to complete precingulum in Thylacinus, and then reduced to total loss in clade K. Chartacter 20 (0 to 1): Loss of metaconule on M1 to M3. Reversed in Thylacinus and then lost again in clade K. Character 33 (1 to 2): Metaconid of M1 entirely lost. Convergent in B. turnbulli. Character 34 (0 to 1)*: Reduction of metaconids on M2 to M4 to minute cuspules. Character 40 (0 to 1)*: Loss of postcingulid on M1 to M3.

Clade H

Content. Wabulacinus ridei, Thylacinus macknessi, Thylacinus potens, Thylacinus megiriani and Thylacinus cynocephalus.

Unambiguous synapomorphies. Character 12 (0 to 1)*: preparacrista on M1 is angled anterobuccally. Character 16 (1 to 2)*: Centrocrista of M1 is straight and parallel with anterior–posterior axis of the tooth. Character 17 (0 to 1)*: Wide, obtuse angle between the postparacrista and premetacrista of M2 and M3, creating a nearly straight centrocrista. Character 19 (0 to 1): Loss of protoconule on upper molars. Convergent in Nimbacinus richi. Character 23 (0 to 1): Stylar cusp B on M1 is highly reduced to absent. Convergent in Badjcinus turnbulli. Character 25 (0 to 1): Loss of stylar cusp C on M2 and M3. Convergent in Muribacinus gadiyuli.

Ambiguous synapomorphies under deltran optimisation. Character 40 (0 to 1)*: Loss of postcingulid on M1 to M3. Character 34 (0 to 1)*: Reduction of metaconids on M2 to M4 to minute cuspules.

Clade I. Thylacinus

Unambiguous synapomorphy. Character 36 (0 to 1) presence of a carnassial notch in the cristid obliqua. Convergent in Ngamalacinus timmulvaneyi.

Ambiguous synapomorphies under deltran optimisation. Character 33 (1 to 2): Complete loss of metaconid on M1. Convergent in Badjcinus turnbulli.

Ambiguous synapomorphies under acctran optimisation. Character 1 (0 to 1): Maxilla-jugal suture passes very close to the margin of the infraorbital foramen so that only a thin sliver of the maxilla separates the jugal from the foramen. Convergent in Ngamalacinus timmulvaneyi. Character 2 (0 to 1): Infraorbital foramen shifted posteriorly to a position dorsal to M2. Convergent in Muribacinus gadiyuli and Ngamalacinus timmulvaneyi. Character 10 (1 to 0): Presence of a complete precingulum on M1. Reversal of a character that was incompletely lost at clade G. Character 17 (1 to 2)*: Straight centrocrista on M2 and M3. Character 18 (1 to 2)*: Extreme elongation of the postmetacrista so that it is over 52% of the length of the tooth. Character 20 (1 to 0): Presence of a metaconule on the upper molars. Reversal of a character that was lost at clade G, subsequently lost again in clade K. Character 30 (0 to 1): Presence of a diastema between P2 and P3. Convergent in Thylacinus macknessi, Nimbacinus and Muribacinus gadiyuli.

Clade J

Content. Thylacinus potens, Thylacinus megiriani and Thylacinus cynocephalus.

Unambiguous synapomorphies. Character 15 (0 to 1): Paracone of upper molars significantly reduced in comparison to metacone. Convergent in Tyarrpecinus rothi. Character 34 (1 to 2)*: metaconids of M2 to M4 completely lost. Character 35 (1 to 2)*: anterior end of cristid obliqua extends to the tip of the protoconid. Character 39 (0 to 1): M4 is longer than M3. Convergent in Mutpuracinus archibaldi. Character 42 (0 to 1): Increased body mass, so that it is greater than 15 kg.

Ambiguous synapomorphies under deltran optimisation. Character 1 (0 to 1): Maxilla-jugal suture passes very close to the margin of the infraorbital foramen so that only a thin sliver of the maxilla separates the jugal from the foramen. Convergent in Ngamalacinus timmulvaneyi. Character 17 (1 to 2)*: Straight centrocrista on M2 and M3. Character 18 (1 to 2)*: Extreme elongation of the postmetacrista so that it is over 52% of the length of the tooth.

Clade K

Content. Thylacinus megiriani, Thylacinus cynocephalus.

Unambiguous synapomorphies. Character 10 (0 to 2)*: complete loss of precingulum on M1. Character 11 (0 to 1)*: loss of precingulum on M3. Character 20 (0 to 1): loss of metaconule on upper molars. Convergent in Tyarrpecinus rothi and Wabulacinus ridei (deltran optimisation) or reversal of a character reacquired in Thylacinus (acctran optimisation). Character 21 (0 to 1)*: M3 greater than 5% longer than M2. Character 22 (0 to 1)*: M3 is longer than it is wide. Character 28 (0 to 1)*: loss of stylar crest on M3. Character 32 (0 to 1)*: Presence of a diastema between P3 and M1.

Ambiguous synapomorphies under deltran optimisation. Character 2 (0 to 1): Infraorbital foramen shifted posteriorly to a position dorsal to M2. Convergent in Muribacinus gadiyuli and Ngamalacinus timmulvaneyi. Character 30 (0 to 1): Presence of a diastema between P2 and P3. Convergent in Muribacinus gadiyuli, Nimbacinus and Thylacinus macknessi.

Ambiguous synapomorphies under acctran optimisation. Character 7 (1 to 0). P2 is shorter than M1. Reversal of a character that evolved in clade G. Character 9 (1 to 0). Loss of posterolingual cusp on P3. Reversal of a character that evolved in Thylacinidae.

Additional Information and Declarations

Competing Interests

Author Contributions

The author declares there are no competing interests.

Adam M. Yates conceived and designed the experiments, performed the experiments, analyzed the data, contributed reagents/materials/analysis tools, wrote the paper, prepared figures and/or tables, reviewed drafts of the paper.

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
