# Peer review of "New craniodental remains of Thylacinus potens (Dasyuromorphia: Thylacinidae), a carnivorous marsupial from the late Miocene Alcoota Local Fauna of central Australia"

_PeerJ, doi:10.7717/peerj.547_

## Round 0.1 · original submission · Minor Revisions

- Both reviewers suggest that the source for the dental terminology be clarified. This should be incorporated in the revision. Reviewer 1 in particular suggests some areas of clarification for this in the description, including a check for terminological consistency as well as orientation labels on the figures. Please incorporate this into your revision.
- Reviewer 2 suggests that the title should be revised (in particular, eliminating the "Part 1" phrase), and I agree. An alternative title provided by the reviewer is one possibility.
- Reviewer 2 suggests some rearrangement of the introduction, particularly to place the information on thylacines first. Because thylacines are the focus of the paper, a little more context for them in the introduction would be appropriate. I do think that the fairly detailed locality information should be retained, though, perhaps under a subheading.
- Reviewer 2 recommends rephotography (to improve focus in some images [e.g., Figure 8] as well as accommodate color variations on the specimens). This is highly desirable in my opinion, but if it is not possible, I feel that the interpretive drawings do convey much of the necessary information.
- Both reviewers request some justification for Th. potens being "an unusually variable species." Please address this in revision.
- Please address the other, more minor, suggestions from the reviewers in your revision and/or response letter.

Other notes not discussed by the reviewers:
- For the mass estimates, I would include a translation of what the percentage error estimation means in kilograms, rather than just percentages. Is the error estimation a 95% confidence interval? What does it mean? What is a smearing estimate? These terms are not at all defined in the paper otherwise, but should be.
- For the phylogenetic analysis, I would use a single taxon instead of the composite "Dasyuridae". Current best practice in phylogenetic analysis is to avoid use of composite taxa.

·

Basic reporting

N/A

Experimental design

N/A

Validity of the findings

N/A

Additional comments

A good paper worth publishing but with some minor revisions required.

TITLE
The ‘Part 1’ bit adds text and sounds pretentious (like starting a paper title with ‘On’, e.g., “On the importance of my research, Part 1”). Not only that, it adds redundancy for much of the title for the next papers in the would-be “series”. Also, I can think of several cases of orphaned ‘Part 1’ papers.

What is wrong with “New craniodental remains of Thylacinus potens (Dasyuromorphia: Thylacinidae), a carnivorous marsupial from the late Miocene Alcoota Local Fauna of central Australia”?

INTRO
The paper is mainly focused on describing new thylacine material, but most of the Intro is filled with details on the site from which the specimens came at the expense of more information on thylacines. The relevance of much of this is questionable in the context of a fundamentally taxonomic paper. Certainly, the Intro should lead off talking about thylacines not the history of the Alcoota site including relocating quarries which is way off-topic! I believe much of it belongs in a separate site paper. When thylacines are introduced it is without any background information on what they are. Knowledge is assumed. Really, the Intro needs to begin by introducing thylacines and what we know about them, recent and fossils, and keep the site stuff to a minimum, possibly under a separate subheading.

lines 47-48 - Woodburne’s excavation was 52 years ago.

METHODS
Terminology
The info presented is unnecessary / not relevant. Regarding this point, all the author needs to say is that serial designation of the cheek dentition follows Flower (1867) and Luckett (1993). However, I would have thought there is a bit more to terminology than tooth numbering, so give some more info on this and cite the references from which feature terminology has been derived.

line 64 - body mass not bodymass

line 100 - two parts not 2 parts

RESULTS
Description very good overall, although I must confess that I did not check every element of the description against the images and tabulated numerical data provided.

I’m not sure that I agree though with the comment (lines 120-124) that “The snout of Th. potens was probably not proportionately shorter or deeper than that of Th. cynocephalus and that the crowding of the premolar teeth seen in this species is more likely to be the result of relative enlargement of these teeth as opposed to relative shortening of the jaw.” Unless the maxillary fragments are misaligned with the premaxillary portion (which is precisely what Murray did initially with the reconstruction of T. megiriani until picked up by reviewer J. Muirhead - see Murray 1997) then the snout is clearly short compared with T. cynocephalus. Not that the average Joe reading this paper would be able to tell, because there are now figures of T. cynocephalus for comparison. It would be good if there were.

DISCUSSION
I agree that it is likely that these specimens represent a variable T. potens, but what is the basis (line 485) for claiming that it “is an unusually variable species”? Compared with what? High intraspecific variation in such attributes is common in many mammal species.

Similarly, the comment that extensive dental wear in some specimens is “strongly suggestive of durophagy, quite possibly bone-cracking” requires backing up by demonstration that this is NOT the case in T. cynocephalus, which is the only thylacine represented by a significant number of specimens and is most certainly not renowned for its durophagy. On the other hand, T. potens does look to my eye to have had a significantly shorter snout than T. cynocephalus (I would like to see a side-by-side comparison), which would be suggestive of a capacity to produce larger bite forces.

GENERAL WRITING
Expression can be tightened up here and there.

FIGURES
Generally these are good but a) some are blurred or lack adequate field depth, b) occlusal views of dentition really should be in stereo pairs, and c) the teeth of these specimens are mottled similar to the bone which means it is very difficult to discern key features on the photos. Why not coat them in ammonium chloride and then rephotograph? It would help a lot.

·

Basic reporting

No comments

Experimental design

p4. Terminology. Please state what tooth cusp nomenclature and what cranial terminology is being followed in the descriptions.
p4. Size estimation. The author uses regression equations from Myers (2001). Please state the table number of the restricted dasyuromorphian dataset used from Myers (2001) (i.e., as you have for Wroe [2001] and Woodburne [1967] in the next paragraph) e.g., Myers (2001, table x).
p5-6.Cladistic Analysis. The author uses the method of Safe Taxonomic Reduction (STR) by removing Ma. muirheadae aposteriori from the most parsimonious trees generated by the analysis, in order to obtain a ‘reduced consensus tree’. However, no mention of this method for pruning trees is given in the cladistic methodology. It is first mentioned in the Discussion. Please indicate the use of STR in the Cladistic Analysis section of the Methods and provide a brief justification for its use.
Line 738. I would question the validity of using ‘body size’ as a character in a phylogenetic analysis. Body size is influenced by numerous environmental and biological factors. Not only has it been shown to be correlated with both geography (e.g. Bergmann’s rule) and time (e.g. Cope’s rule) it is also affected by resource availability, resource distribution, habitat structure, temperature, rainfall, and interspecific competition. Further, how can the author account for body size differences related to sexual dimorphism when determining character polarities? And what is the justification for ordering such a character?
p6. Systematic Palaeontology. Please include details of the Holotype specimen and Type locality for Thylacinus potens under the sub-headings ‘Holotype’ and ‘Type Locality’. The later could be relatively brief because much of this information is included in the Introduction.
p6-13. Description. I find the authors use of positional terms such as mesial, distalolingual, buccomesial etc in the description quite confusing. Most descriptions of thylacinid dentitions and marsupial dentitions in general, use the terms ‘anterior’, ‘posterior’, ‘buccal’ (or labial), ‘lingual’, ‘dorsal’ and ‘ventral’ to describe positional relationships of cusps/crests on teeth. In this description, these terms and the terms ‘mesial’ and ‘distal’ as well as combinations of all of these terms and ‘downward’ and ‘horizontal’ are used. I would like to see the description more comparable with other published descriptions of thylacinid dentitions, or at least, I would like to see orientation labels (e.g., buccal, lingual, anterior, posterior, mesial, distal) on the figures of the dentitions to more clearly define what the author means by these terms. The term ‘mesial’ is usually used to refer to a feature that is towards the midline of a structure, but that does not appear to be the case here. Sometimes it appears to be used in place of the term ‘anterior’. Further, generally the terms ‘posterior’ and ‘distal’ refer to the same position/orientation when describing positions on teeth, but they are used in the same sentence in the descriptions given here. For example, lines 321-322 (“Posterior to the paraconid is a mesiodistally short and buccolingually broad talonid shelf”). Similarly, both ‘ventrally’ and ‘downwards’ are used but I would think they mean the same thing. For example, in lines 213-214 (“It curves downwards and buccoventrally from the distolingual edge of the metacone terminating at the distalobuccal corner of the tooth…”), the term ‘downwards’ could be replaced by ‘ventrally’. Further there is random spelling and use of the terms distolingual, distalolingual, linguodistal, distobuccal, distalobuccal, mesial-distal, mesiodistal etc.

Validity of the findings

Line 485. The author states that “Th. potens is an unusually variable species”. On what is this assessment based? i.e., is it unusually variable relative to other thylacinid species or other marsupial taxa? I would like to see a short paragraph discussing potential morphological and/or size variation for Th. cynocephalus. Ride (1964) indicates that Th. cynocephalus was highly sexually dimorphic. Was this reflected in the metric data used in this analysis and does that imply potential sexual dimorphism in the larger Th. potens? Have previous authors such as Woodburne (1967) and Dawson (1982) discussed qualitative or quantitative variation in Th. cynocephalus dentitions to qualify this statement?

Additional comments

Size of Thylacinus potens and palaeobiology. It would be interesting to speculate on niche partitioning and potential prey-size preference for Th. potens. In the introduction the author mentions that Th. potens was sympatric with two other carnivores in the Alcoota LF: Ty. rothi (body weight 5 kg) and Wakaleo alcootaensis (weight not given). It would be of interest to provide the weight of Wakaleo alcootaensis if possible, to see whether it was in the same range as Th. potens and discuss what that may suggest about niche partitioning among carnivores at Alcoota. I refer the author to a recent study published on the prey size preference of Nimbacinus dicksoni.
Attard MRG, Parr WCH, Wilson LAB, Archer M, Hand SJ, et al. (2014) Virtual Reconstruction and Prey Size Preference in the Mid Cenozoic Thylacinid, Nimbacinus dicksoni (Thylacinidae, Marsupialia). PLoS ONE 9(4): e93088. doi:10.1371/journal.pone.0093088.
Although a biomechanical analysis is beyond the scope of this paper, can the author speculate about the potential prey preference of Th. potens’ based on the results of Attard et al.’s study?

In addition to the above comments there are numerous typo's and inconsistencies throughout the document that I have highlighted in the attached pdf file.

The figures (photographs and illustrations) are nicely presented.

---

## Round 0.2 · accepted · Accept

Thank you for your close attention to the comments from the reviewers. In my opinion, you have sufficiently addressed their concerns.